

Estimation of Liquid Water Path in Stratiform Precipitation Systems using Radar Measurements
during MC3E
Jingjing Tian[1], Xiquan Dong[1], Baike Xi[1], Christopher R. Williams[2], and Peng Wu[1]
[1]Department of Hydrology and Atmospheric Sciences, University of Arizona, Tucson, Arizona,
USA
[2] Department of Ann and H.J. Smead Aerospace Engineering Sciences, University of Colorado
Boulder
Manuscript Submitted to Atmospheric Measurement Techniques
Corresponding author address: Dr. Xiquan Dong, The Department of Hydrology and
Atmospheric Sciences, University of Arizona, 1133 E. James Rogers Way, Tucson, AZ 85721-

18  0011.

Email: xdong@email.arizona.edu; Phone: 520-621-4652





**Abstract**
In this study, the liquid water path (LWP) in stratiform precipitation systems is retrieved, which
is a combination of rain liquid water path (RLWP) and cloud liquid water path (CLWP). The
retrieval algorithm uses measurements from the vertically pointing radars (VPRs) at 35 GHz and
3 GHz operated by the U.S Department of Energy Atmospheric Radiation Measurement (ARM)
and National Oceanic and Atmospheric Administration (NOAA) during the field campaign
Midlatitude Continental Convective Clouds Experiment (MC3E). The measured radar
reflectivity and mean Doppler velocity from both VPRs and spectrum width from the 35 GHz
radar are utilized. With the aid of the cloud base detected by ceilometer, the LWP in the liquid
layer is retrieved under two different situations: (I) no cloud exists below the melting base, and
(II) cloud exists below the melting base. In (I), LWP is primarily contributed from raindrops
only, i.e., RLWP, which is estimated by analyzing the Doppler velocity differences between two
VPRs. In (II), cloud particles and raindrops coexist below the melting base. The CLWP is
estimated using a modified attenuation-based algorithm. Two stratiform precipitation cases (20
May 2011 and 11 May 2011) during MC3E are illustrated for two situations, respectively. With
a total of 14 hours of samples during MC3E, statistical results show that the occurrence of cloud
particles below the melting base is low (8%), however, the mean CLWP value can be up to 0.87
kg m$^{-2}$, which is much larger than the RLWP (0.22 kg m$^{-2}$). When only raindrops exist below the
melting base, the averaged RLWP value is larger (0.33 kg m$^{-2}$) than the with cloud situation. The
overall mean LWP below the melting base is 0.39 kg m$^{-2}$ for stratiform systems during MC3E.




## 1. Introduction

Clouds in stratiform precipitation systems are important to the Earth's radiation budget.

The vertical distributions of cloud microphysics, ice and liquid water content (IWC/LWC),
determine the surface and top-of-the-atmosphere radiation budget and redistribute energy in the
atmosphere (Feng et al., 2011; 2018). Also, stratiform precipitation systems are responsible for
most tropical and midlatitude precipitation during summer (Xu, 2013). However, the
representation of those systems in global climate and cloud-resolving models are still challenging
(Fan et al., 2015). One of the challenges is due to the lack of comprehensive observations and
retrievals of cloud microphysics (e.g. prognostic variables IWC and LWC) in stratiform
precipitation systems. Liquid water path (LWP), defined as an integral of LWC in the
atmosphere. It is a parameter used to provide the characterization of liquid hydrometeors in the
vertical column of atmosphere and study clouds and precipitation. The estimation of LWC/LWP
is one of the critical objectives of the US Department of Energy's (DOE) Atmospheric Radiation
Measurement (ARM) Program (Ackerman and Stokes, 2003).

LWP can be retrieved using the ground-based MicroWave Radiometer (MWR) sensed

downwelling radiant energy at 23.8 and 31.4 GHz (Liljegreen et al., 2001). In last two decades,
ARM has been operating a network of 2-channel (23.8- and 31.4-GHz) ground-based MWR to
provide a time series of LWP at the ARM Southern Great Plains (SGP) site (Cadeddu et al.,
2013). Absorption-based algorithms using multichannels of MWRs have been widely used to
retrieve cloud LWP (e.g., Liljegren et al. 2001; Turner, 2007), and it is known to be accurate
methods to estimate LWP of nonprecipitating clouds with mean LWP error of 15 g m$^{-2}$ (Crewell
and Löhnert, 2003). However, in precipitating conditions, LWP retrieved from conventional





MWR are generally not valid due to the violation of the Rayleigh assumption when large
raindrops exist (e.g., Saavedra et al., 2012). In addition, large increase of brightness
temperatures is measured as a result of the deposition of raindrops on the MWR's radome.
Unfortunately, it is very hard to model and quantify this increase from rain layer on the radome
(Cadeddu et al., 2017). This "wet-radome" issue largely inhibits the retrieving of LWPs using
ground-based MWR during precipitation. Due to the limitations of retrieving LWP from MWR
during precipitation, cloud and precipitation radars were used to simultaneously retrieve LWP
(Matrosov, 2010).
In the precipitating system, the liquid water cloud droplets and raindrops often coexist in
the same atmospheric layer (e.g., Dubrovina, 1982; Mazin, 1989; Matrosov, 2009, 2010),
indicating that the LWP consists of both cloud liquid water path (CLWP) and rain liquid water
path (RLWP). However, the discrimination between suspended small cloud liquid water droplets
and precipitating large raindrops is a very challenging remote sensing problem. Even though the
partitioning of LWP into CLWP and RLWP is important in cloud modeling (Wentz and Spencer,
1998; Hillburn and Wentz, 2008), there are few studies retrieved RLWP and CLWP
simultaneously and separately (Saavedra et al., 2012; Cadeddu et al., 2017). Battaglia et al.
(2009) developed an algorithm to retrieve RLWP and CLWP from the six Advanced Microwave
Radiometer for Rain Identification (ADMIRARI) observables under rainy conditions. Saavedra
et al (2012) developed an algorithm using both ADMIRARI and a micro rain radar to retrieve
and analyze the CLWP and RLWP for midlatitude precipitation during fall. In addition to these
RLWP and CLWP estimations mainly from passive microwave radiometers, there are several
studies to estimate the LWP using active radar measurements only. Ellis and Vivekanandan
(2011) developed an attenuation-based technique to estimate LWC, which is the sum of cloud



water contents (CLWC) and rain liquid water contents (RLWC), using simultaneous S- and Ka-
band scanning radars measurements.    However, it is not always applicable of using these
techniques to retrieve LWC.  If raindrop diameters are comparable to at least one of the radars'
wavelength, "Mie effect" will be included in the measured differential reflectivity, however this
"Mie effect" is not very distinguishable from differential attenuation effects (Tridon et al., 2013;
Tridon and Battaglia 2015).

Matrosov (2009) developed an algorithm to simultaneously retrieve CLWP and layer-

mean rain rate using the radar reflectivity measurements from three ground-based W-, Ka-, S-
bands radars.  The CLWP were retrieved based on estimating the attenuation of cloud radar
signals compared to S-band radar measurements.  Matrosov (2010) developed an algorithm to
estimate CLWP using a vertical pointing Ka-band radar and a nearby scanning C-band radar.
The layer-mean rain rate was first estimated with the aid of surface disdrometer, and then CLWP
was retrieved by subtracting the rain attenuation from total attenuation measured from two radars.
For the estimation of RLWP, Williams et al. (2016) developed a retrieval algorithm for rain drop
size distribution (DSD) using doppler spectrum moments observed from two collocated vertical
pointing radar (VPRs) at frequencies of 3 GHz and 35 GHz.  The retrieved air motion and DSD
parameters were evaluated using the retrievals from a collocated 448-MHz VPR.

In this study, the CLWP retrieval algorithms in Matrosov (2009 and 2010) have been

modified given the available radar measurements, vertical pointing Ka- and S-band radars,
during the Midlatitude Continental Convective Clouds Experiment (MC3E) field campaign.  For
the estimation of RLWP, we will basically follow the idea described in Williams et al. (2016) to
retrieve microphysical properties for raindrops, however instead of retrieving vertical air motion
and rain DSDs (Williams et al., 2016), this study aims at retrieving RLWCs, and then integrating





RLWCs over the liquid layer to estimate RLWP. Overall, in this study, algorithms from three
former publications are modified and combined to estimate the LWP in the stratiform
precipitating systems.
The goals of this study are to retrieve the LWP, which includes both RLWP and CLWP
retrievals using radars measurements, and tentatively answer two questions based on
observations and retrievals in the stratiform precipitation systems during MC3E: (1) what is the
occurrence of cloud below the melting base in the stratiform precipitation systems; (2) what are
the values of simultaneous CLWP, RLWP and LWP, and how does CLWP or RLWP contribute
to the LWP. Note that the CLWP and RLWP are constrained in a stratiform precipitation layer
below the melting base and above the surface. The LWP estimations in this study are primarily
aimed at stratiform precipitating events exhibiting melting-layer features from radar
measurements with lower-to-moderate rain rates (RR < 10 mm hr$^{-1}$). The instruments and data
used in this study are introduced in section 2. Section 3 describes the methods of retrieving LWP
(both RLWP and CLWP). Section 4 illustrates two examples and followed by statistical results
from more samples during MC3E. The last section gives the summary and conclusions.
Acronyms and abbreviations are listed in Table 1.

**2. Data**
The MC3E field campaign, co-sponsored by the NASA Global Precipitation
Measurement and the U.S. DOE ARM programs, was conducted at the ARM SGP (northern
Oklahoma) during April-June 2011 to study convective clouds and improve model
parametrization (Jensen et al., 2015). MC3E provided an opportunity to develop new retrieval
methods to estimate cloud microphysics and precipitation properties in precipitation systems



(Giangrande et al., 2014; Williams, 2016; Tian et al., 2016; Tian et al, 2018). Several stratiform
rain cases were observed by the VPRs during MC3E (as shown in Fig. 1). Distinct signatures of
"bright banding" are detected from VPRs. To retrieve LWP associated with stratiform
precipitation, this study mainly uses the observations from two co-located VPRs operating at 3-
GHz and 35-GHz at DOE ARM SGP Climate Research Facility.
**2.1 Vertical Pointing Radars**

The 3-GHz (S-band) VPR was deployed by NOAA Earth System Research Laboratory

for the six-weeks during the MC3E. The NOAA 3-GHz VPR is a vertical pointing radar with
2.6° beamwidth monitoring precipitation overhead. This 3-GHz profiler bridges the gap between
cloud radars, which are used to provide the structure of nonprecipitating clouds but are severely
attenuated by rainfall, and precipitation radars, which, although unattenuated by rainfall,
generally lack the sensitivity to detect more detailed cloud structure. The 3-GHz VPR observes
the raindrops within the Rayleigh scattering regime and its signal attenuation are negligible
through the rain. The temporal resolution of the profiles of Doppler velocity spectra is 7 seconds
and the vertical resolution is 60 meters. The 3-GHz VPR operated in two modes: a precipitation
mode and a low-sensitivity mode. The precipitation mode observations are used in this study.

The Ka-band ARM zenith radar (KAZR) is also a vertical pointing radar, operating at 35

GHz permanently deployed by DOE ARM at the SGP site. The KAZR measurements include
reflectivity, vertical velocity, and spectral width from near-ground to 20 km. The KAZR data
used in this study are the KAZR Active Remote Sensing of Clouds (ARSCL) product produced
by the ARM (www.arm.gov). The KAZR-ARSCL corrects for atmospheric gases attenuation
and velocity aliasing. By selecting the mode with the highest signal-to-noise ratio at a given
point, data from two simultaneous operating modes (general and cirrus mode) are combined for



each profile to provide the "best estimates" of radar moments in the time-height fields. The
vertical and temporal resolutions of KAZR-ARSCL product are 30 meters and 4 seconds,
respectively. Since the 3-GHz and 35-GHz VPRs are independent radars with different dwell
time and sample volumes (Williams et al., 2016), the radar observations are processed to 1-min
temporal and 60-m vertical resolutions in this study.
**2.2 Disdrometers**
DOE ARM program maintains a suite of surface precipitation disdrometers.
Measurements and estimations from the Distromet model RD-80 disdrometer and NASA two-
dimensional video disdrometers (2DVD) deployed at the ARM SGP site are used in this study.
The RD-80 disdrometer provides the most continuous raindrop size distribution (DSD)
measurements at high spectral (20 size bins from 0.3 to 5.4 mm) and temporal resolutions (1
minute), and its minimal detectable precipitation amount is 0.006 mm hr$^{-1}$. From 2DVD, the rain
DSDs are observed from 41 bins (0.1 - 10 mm), and its minimal detectable precipitation amount
is 0.01 mm hr$^{-1}$. In addition to rain rate, the mean mass-weighted raindrop diameter ($D_m$) is also
provided from 2DVD, which is used for evaluating retrieved $D_m$ from radar measurements.
**2.3 Ceilometer**
A Vaisala laser ceilometer (CEIL) operates at the SGP Central Facility, sensing cloud
presence up to a height of 7700m with 10-m vertical resolution. The laser ceilometer transmits
near-infrared pulses of light, and the receiver detects the light scattered back by clouds and
precipitation. It is designed to measure cloud-base height.

**3. The Methodology of Liquid Water Path Estimation**
As mentioned earlier, both RLWP and CLWP contribute to the LWP. With aid of the
cloud base height detected by ceilometer, LWP is retrieved under two different situations: (I) the




cloud base is higher than the melting base and (II) the cloud base is lower than the melting base.
For situation (I), there are almost no cloud droplets below melting base (CLWP = 0), and thus
the LWP below the melting base is solely from raindrops. The LWP is calculated by integrating
RLWCs over this layer. The RLWCs could be retrieved by analyzing the measured Doppler
Velocity Differences ("*DVD Algorithm*") from two collocated VPRs. In situation (II), the small
cloud droplets and large raindrops coexist below the melting base. Both raindrops and cloud
particles contribute to LWP. RLWP will be still estimated using "*DVD Algorithm*". CLWP will
be retrieved using an attenuation-based algorithm named as "*Attenuation Algorithm*". The
algorithms for LWP estimation are summarized in a flowchart (Fig. 2).
**3.1 Situation I (no cloud droplets exist below the melting base)**

The algorithm from Williams et al. (2016) was developed based on an assumption that

the 3-GHz VPR operates within the Rayleigh scattering regime for all raindrops, while the 35-
GHz VPR operates within the Rayleigh scattering regime for small raindrops (diameters < ~1.3
mm) and non-Rayleigh scattering regime for larger raindrops (diameters ≥ ~1.3 mm). The
different scattering regimes for the two operating frequencies result in different estimated radar
moments. These estimated radar moments are in functions of rain microphysics. Thus, the rain
microphysics could be retrieved with given measured radar moments. The details of this "*DVD*
*Algorithm*" and uncertainty estimation are introduced in Appendix A.
**3.2 Situation II (cloud particles and rain droplets coexist below the melting base)**

In situation (II), substantial cloud particles exist below melting base, and both RLWP and

CLWP retrievals are needed to estimate the LWP. The total two-way attenuation of 35-GHz
VPR signals, A (in decibels, dB), in a layer between the melting base and the cloud base, mainly





consists of rain attenuation, liquid clouds attenuation, and gaseous attenuation. The total
attenuation (A) are expressed as:
$$A = 2\ C\ R_m\ \Delta H + 2\ B\ CLWP + G. \qquad (1)$$
$R_m$ is layer-mean rain rate, and $\Delta H$ (km) is the thickness of the layer (Matrosov, 2009). G is the
two-way attenuation/absorption from atmospheric gases, which is relatively small, and the
absorption by gases has been already corrected in the KAZR ARSCL dataset and is assumed to
be zero in our retrieval.
C and B are the coefficients for rainfall and cloud liquid water attenuation.
$$B = 0.0026\pi\lambda^{-1}\mathrm{Im}[-(m^2-1)(m^2+2)^{-1}], \qquad (2)$$
where $\lambda$ is the wavelength of Ka-band radar, and m is the complex refractive index of water.
The unit of B is dB/g m$^{-2}$.
$$C = 0.27\ b, \qquad (3)$$
where b is the correction factor considering raindrop fall velocities with changing air density.
$$b = (\rho_{am}/\rho_{a0})^{0.45}, \qquad (4)$$
where $\rho_{am}$ and $\rho_{a0}$ are the mean air density in the rain layer and the density at normal atmospheric
conditions.
Based on (1), CLWP can be written as:
$$CLWP = \frac{A - 2\ C\ R_m\ \Delta H - G}{2\ B} \qquad (5)$$
The attenuation (A) is estimated by comparing the drop in Ka-band reflectivity with the
un-attenuated S-band reflectivity through the cloud. Assuming the changes in reflectivity with
altitude due to changes in raindrop size distributions with altitude are similar for Ka- and S-band
reflectivities, then the difference in reflectivities through the cloud is a proxy for attenuation.
This can be expressed using





$$A \cong [Z_{Ka}(cloud\ base) - Z_{Ka}(melting\ base)] - [Z_S(cloud\ base) - Z_S(melting\ base)] \quad (6)$$
Notice that the absolute calibration of the radar was not important to the retrieval results since
the retrieval of CLWP used S-Ka differential attenuation. This avoids the radar calibration
(Tridon et al., 2015 and 2017), which is a serious issue limits the accuracy of radar retrievals.
The $R_m$ is estimated as:
$$R_m = \frac{\sum_{CB}^{MB} RR(h) \times \Delta h}{\Delta H}, \quad (7)$$
where $\Delta h$ equals 60 meters and MB and CB are the melting base and the cloud base. RRs in the
layer between the melting base and the cloud base are calculated from the "*DVD algorithm*".
The uncertainties of retrieved CLWP are mainly due to the uncertainties of estimated $R_m$
and observed total attenuation from VPRs. The value of $B_k$ is on the order of 1 dB/kg m$^{-2}$. The
uncertainty of retrieved CLWP would be ~ 0.25 kg m$^{-2}$ with 0.5 dB uncertainty from measured
radar reflectivity difference or ~ 0.5 kg m$^{-2}$ for 1.0 mm hr$^{-1}$ uncertainty from estimated layer-
mean rain rate. Compared to the typical mean rain rate observed in the stratiform system (~ 2 - 4
mm hr$^{-1}$), 1.0 mm hr$^{-1}$ represents a ~ 30% uncertainty. The uncertainty for CLWP retrievals is
roughly estimated as ~ 0.56 kg m$^{-2}$ (sqrt (0.25$^2$+0.5$^2$)) in this study. For reference, the expected
uncertainty is reported as ~ 0.25 kg m$^{-2}$ for typical rainfall rates (~ 3 - 4 mm hr$^{-1}$) in Matrosov
(2009) retrieval method.

**4. Retrieval Results and Discussions**
**4.1 Case Studies**
Even though situation (I) is dominated (Fig. 1), especially in Case A, the ceilometer
cloud base estimates can be lower than the melting base (Cases B to D). Two case studies (20





May 2011 and 11 May 2011) are given as examples to demonstrate the estimation of LWP in
stratiform precipitation system for two different situations.
**4.1.1 Case A**

On 20 May 2011, an upper level low-pressure system at central Great Basin moved into

the central and northern Plains, while a surface low pressure at southeastern Colorado brought
the warm and moist air from the southern Plains to a warm front over Kansas. and a dry line
extended southward from the Texas-Oklahoma. With those favorable conditions, a strong north-
south oriented squall line developed over Great Plains and propagated eastward. The convection
along the leading edge of this intense squall line exited the ARM SGP network around 11 UTC
20 May leaving behind a large area of stratiform rain (Case A in Fig. 1). This stratiform system
passed over the ARM SGP site and observed by two VPRs, and disdrometers as shown in
Figures 1a-1c. It clearly shows the 3-GHz radar echo tops are much lower than those from the
35 GHz VPR. Even though there is attenuation at 35-GHz by the raindrops and melting
hydrometeors, the 35-GHz radar can still detect more small ice particles at near the cloud top.
The "bright band", which occurs in a uniform stratiform rain region, is clearly seen from the 3-
GHz VPR (a sudden increase and then decrease in radar reflectivity) but is not obvious from the
35-GHz VPR due to the non-Rayleigh scattering effects at 35 GHz (Sassen et al., 2005;
Matrosov, 2008).

Figures 1a-1b clearly show that the ceilometer detected cloud base is in the middle of the

melting layer, indicating almost no cloud particles below the melting layer and the LWP in the
liquid layer equals to RLWP. The RLWP is retrieved using the "*DVD Algorithm*" introduced in
section 3.1 and Appendix A. Figure 3 shows an example of the DVD retrieval algorithm at
13:40 UTC on May 20, 2011. Radar reflectivity from 3 GHz, Doppler velocities from 3 GHz



and 35 GHz, and spectrum variance from 35 GHz are the inputs of DVD algorithm. The
Doppler velocity differences (3 GHz – 35 GHz) from the surface to 4 km are also plotted in Fig.
3d. The melting base is defined as the height of maximum curvature in the radar reflectivity
profile at 3 GHz (Fabry and Zawadzki, 1995), which is clearly seen at 2.5 km in Fig. 3. Below
2.5 km, the Doppler velocity differences between the two VPRs become relatively uniform,
indicating that the process of melting snow/ice particles into raindrops is completed. Retrieved
profiles of rain microphysical properties and their corresponding uncertainties (horizontal bars at
different levels) in the rain layer (0 – 2.5 km) are shown in Figs 3f-3h. In general, the retrieved
$D_m$ values from the surface to 2.5 km are nearly a constant of ~2 mm (Fig. 3f), while the
retrieved RLWC and rain rate values slightly decrease from 2.5 km to the surface. One of the
highlights of this study is, in addition to the surface rain rate, which can usually be observed
using surface disdrometers, the vertical profiles of rain microphysical properties are retrieved.
These retrieved rain microphysical properties will shed light on the understanding of liquid cloud
and rain microphysical processes (like condensation, evaporation, autoconversion and accretion
etc.) in the models.

To evaluate the rain property retrievals, we compare the retrieved rain microphysical

properties, the $D_m$, and rain rate at the surface, with the surface disdrometers measurements (Fig.
4). The $D_m$ values range from 1.0 to 2.5 mm during a 3.5-hr period with nearly identical mean
values of 1.79 mm and 1.81 mm from both retrievals and 2DVD measurements. There are large
variations for rain rates, ranging from 0 to 8 mm hr$^{-1}$, with means of 3.19, 3.17 and 2.88 mm hr$^{-1}$,
respectively, from 2DVD, RD-80 and radar retrieval. The mean rain rates from 2DVD and RD-
80 measurements are almost the same although there are relatedly large differences during
certain time periods, while the retrievals from this study, on average, underestimate the rain rate



by ~10% compared to the disdrometer measurements. More statistics (mean differences, their 95%
confidence intervals of mean differences and root mean square errors) can be found in Table 2.
Overall, the mean differences are within the retrieval uncertainties. The variation of RLWP (Fig.
4c) mimics the variation of retrieved rain rate in Fig. 4d. The mean value of RLWP is 0.56 kg m$^{-2}$
for this case, which is also the LWP below the melting base.

**4.1.2 Case B**

On 11 May 2011, a surface cold front moved across the Oklahoma-Texas area and then
convections were initiated. At 1600 UTC, a mesoscale convective system organized with a
parallel stratiform precipitation region. Two-three hours later (~1830 UTC), the mesoscale
convective system was transitioned to a trailing stratiform mode passed over the ARM SGP site.
The large stratiform regions are observed by two VPRs and disdrometers as shown in Figs 1d-1f.
Figures 1d-1f clearly show that the ceilometer detected cloud bases are lower than the melting
bases occasionally. Under this situation, both RLWP and CLWP could contribute to the LWP
below the melting base.
Firstly, the surface rain microphysics ($D_m$, RLWC, rain rate and RLWP) are retrieved
using "*DVD Algorithm*". These rain property retrievals are compared with the surface
disdrometers measurements (Fig. 5). The $D_m$ values at the surface range from 0.8 to 2.2 mm
during a 4.5-hr period with the mean values of 1.46 mm and 1.57 mm, respectively, from both
retrievals and 2DVD measurements. The difference between the retrieval and 2DVD
measurement may be due to different sampling volumes between radar and the surface
disdrometer, as well as wind shear. To further investigate the difference, the measurements from
five NASA 2DVDs located within 5 km away from VPRs are collected and processed. The
almost same mean values and slight variation from 5 NASA 2DVDs measurements suggest that





the difference between radar retrievals and the surface disdrometer measurements may be true,
while averaging from more measurements can only smooth the variation.

The mean rain rate values from five NASA 2DVDs and the surface disdrometer are very

comparable, with a mean difference of 0.3 mm hr$^{-1}$. The almost same mean values between the
surface disdrometer and 5 NASA 2DVDs measurements suggest that the DVDs apart within 5
km can capture very similar rain properties during a longer time period, such as 4.5 hours in this
case, although there are some large differences from their point-to-point measurements. The rain
rates, in this case, vary quite large, ranging from 0 to 9 mm hr$^{-1}$ with means of 1.81, 1.64 and
1.98 mm hr$^{-1}$, respectively from single 2DVD, RD-80, and our retrieval. It is found that, from
both Case A and Case B, the mean value from RD-80 is smaller than that from 2DVD. This may
be due to the different ranges of measurable drop sizes from two types of disdrometers (0.3 - 5.4
mm for RD 80, while 0.1 to 10 mm for 2DVD). More statistics can be also found in Table 2.
Overall, the mean differences are still within the retrieval uncertainties for this case.

Secondly, the CLWP is retrieved using "Attenuation Algorithm" introduced in section

3.2. Figure 5c shows the time series of RLWP, CLWP and LWP retrievals. It is found that the
CLWP values (when they exist) are usually larger than RLWP values in the same vertical
column. When cloud droplets and raindrops coexist below the melting base, the mean values are
0.31 kg m$^{-2}$ and 1.00 kg m$^{-2}$ for RLWP and CLWP, and the corresponding LWP below the
melting layer is 1.31 kg m$^{-2}$. While when only raindrops exist below the melting base, there is no
CLWP (CLWP =0), and the RLWP and LWP are the same (with average of 0.33 kg m$^{-2}$). It is
noticed that even though the occurrence of CLWP is low (12%) in this case, the value of CLWP
can be very large when it exists, and it is about two times larger than the mean RLWP. The
mean value of LWP is 0.45 kg m$^{-2}$ for all the sample in Fig. 5c.



### 4.2 Statistical Results

Box and whisker plots of retrieved RLWP, CLWP and LWP for situations (I), (II) and all samples during MC3E are shown in Fig. 6. The horizontal orange and red dashed lines indicate the median and mean, boundaries of the box represent the first and third quartiles, and the whiskers are the 10$^{th}$- and 90$^{th}$ -percentiles. During MC3E, a total of 14 hours of stratiform rain were observed by VPRs at the ARM SGP Climate Research Facility, in which 92% and 8% the samples are categorized into the situations (I) and (II), respectively. The mean RLWPs are 0.33 kg m$^{-2}$ and 0.22 kg m$^{-2}$ for the situations (I) and (II). There are a substantial amount of small cloud droplets sustaining in the rain layer and have not yet converted to larger raindrops, which may partially explain smaller RLWP in the situation (II). The mean value of surface rain rate is 1.78 mm hr$^{-1}$ when cloud droplets exist, which is also smaller than the mean value (2.06 mm hr$^{-1}$) in the rain-only situation. The mean CLWP in the situation (II) is as large as ~0.87 kg m$^{-2}$ even though their occurrence is very low (8%), which is much larger than mean RLWP in the liquid layer. The ratio of RLWP and CLWP ranges from 4:1 to 2:1 for precipitating shallow marine clouds reported at Lebsock et al. (2011), while our results from MC3E do not seem to have a clear linear relationship between CLWP and RLWP (figure is not shown). The LWP from the situation (II) is much larger than the mean LWP from the situation (I), which is primarily contributed by cloud droplets. The overall mean LWP for stratiform rain during MC3E is 0.39 kg m$^{-2}$.

We also processed the ARM MWR retrieved LWPs during MC3E and compared with our retrievals as illustrated in Fig. 7. Statistical results of the retrieved LWPs from this study and MWR are averaged for each measured rain rate bins (bin size = 0.25 mm hr$^{-1}$). When the rain rate is greater than ~ 6mm hr$^{-1}$, there are no MWR LWP retrievals. Fig. 7b shows that the MWR retrieved LWPs, as expected, monotonically increase with rain rate, which is possible due to the



"wet radome" effect (Cadeddu et al., 2017). "Wet radome" is a particularly complicated
situation because the standing water often looks physically like a layer and less like a collection
of drops, making the MWR overestimate LWPs (personal communication with Dave Turner,
2018), and so far, no effective method was found to solve this problem (Cadeddu et al., 2017).

In addition to the issue from standing water on the radome, the scattering effects due to

raindrops also affect MWR retrievals. Two general retrieval methods are commonly used to
retrieve LWP from the observed brightness temperatures: statistical methods (Liljegren et al.,
2001) and physical retrievals (Turner et al, 2007). No matter which retrieval is used, the
radiative transfer code usually only models the absorptions from atmospheric gases and cloud
liquid water. The scattering effect is not taken into consideration during the retrieval, that is, it is
under the assumption that the brightness temperature is primarily due to the emission of cloud
droplets in the MWR retrieval. Even small drizzle particles still have a scattering effect, which
could contribute higher brightness temperature measured by MWR and result in larger retrieved
LWPs than the "true" LWPs. Therefore, the MWR retrieved LWPs are most likely
overestimated for precipitating clouds.

In this study, we mainly focus on the stratiform rain systems with mean rain rates of 2-4

mm hr$^{-1}$. The scattering effect for large raindrops is more significant than drizzles. Sheppard
(1996) examined the effect of raindrops on MWR brightness temperature measurements at 31
GHz and found that cloud absorption coefficient is only ~2/3 of rain absorption coefficient,
however, the scattering effect of raindrops is not insignificant where its scattering coefficient is
about half of cloud absorption coefficient. Thus, MWR measured brightness temperatures for
precipitating clouds would be higher, due to the scattering by raindrops, than those for non-
precipitating clouds, and then result in higher LWPs than the 'true" LWPs. The differences of



LWPs from MWR and this study are shown in Fig. 7c. The LWP differences increase almost
linearly with increased rain rate. The differences could be due to (1) MWR retrieved LWP
represents the whole vertical column (RWLP and CLWP below melting layer, large water coated
ice particles in the melting layer and supercooled LWCs above the melting layer), while our
retrieval only represent the LWP below the melting base; (2) existing uncertainty in retrieved
LWP from this study (~0.6 kg m$^{-2}$ when includeing CLWP estimates).

**5. Summary and Conclusions**
LWP is a critical parameter for studying clouds, precipitation, and their life cycles. LWP
can be retrieved from microwave radiometer measured brightness temperatures during cloudy
and light precipitation conditions. However, MWR-retrieved LWPs are questionable under
moderate and heavy precipitation conditions due to the "wet radome" and non-Rayleigh
scattering effects caused by large raindrops. LWPs below the melting base in stratiform
precipitation systems are estimated, which include both RLWP and CLWP. The measurements
used in this study are mainly from two VPRs, 35-GHz from ARM and 3-GHz from NOAA
during the MC3E field campaign.
In this study, the microphysical properties of raindrops, such as $D_m$, RLWC (and RLWP),
and RR, are estimated following the method described in Williams et al. (2016) using
measurements from co-located Ka- and S-band radars VPRs. The retrieved rain microphysical
properties are validated by the surface disdrometer measurements. Instead of retrieving vertical
air motion and rain DSDs (Williams et al., 2016), this study aims at retrieving RLWCs and then
integrating RLWCs over the liquid layer to estimate RLWP. The CLWP is retrieved based on



the modifications of the methods in Matrosov (2009 and 2010) with available radar
measurements, vertical pointing Ka- and S-band VPRs, during the MC3E field campaign.

The applicability of retrieval methods is illustrated for two stratiform precipitation cases

(20 May 2011 and 11 May 2011) observed during MC3E. Statistical results from a total of 14
hours samples during MC3E show that the occurrence of cloud droplets below the melting base
is low (8%), while the CLWP value can be up to 0.87 kg m$^{-2}$, which is much larger than the
RLWP (0.22 kg m$^{-2}$). When only raindrops exist below the melting base, the averaged RLWP
value is 0.33 kg m$^{-2}$, which is much larger than the mean RLWP in the cloud droplets and
raindrops coexisted situation.

Reliable retrievals of RLWC and RLWP are critical for model evaluation and

improvement, as RLWC (rain mixing ratio) is an important prognostic variable in weather and
climate models. Furthermore, the retrievals in the whole rain layer would be useful to
understand the microphysical processes (i.e., condensation, evaporation, autoconversion, and
accretion etc.) and have great potential to improve model parametrizations in the future. Overall,
the LWP (CLWP and RLWP) retrievals derived in this study can be used to evaluate the models
that separately predict cloud and precipitation separately, and contribute comprehensive
information to study cloud-to-precipitation transitions.

**Appendix A: Doppler Velocity Differences Algorithm (“*DVD Algorithm*”)**

Retrieving RLWC and other rain microphysical properties (i.e., drop size and rain rate) is

based on the mathematics of DSD radar reflectivity-weighted velocity spectral density $S_{DSD}^{\lambda}$
[(mm$^6$ m$^{-3}$) (m s$^{-1}$)$^{-1}$], which is a product of radar raindrop backscattering cross section $\sigma_b^{\lambda}(D)$
(mm$^2$) and DSD number concentration N$_{DSD}$(D) (mm$^{-1}$ m$^{-3}$):



$$S^\lambda_{DSD}(v_z) = [\frac{\lambda^4}{\pi^5 |K_w|^2} \sigma^\lambda_b] N_{DSD}(D) \frac{dD}{dv_z} .$$
(A1)

The $\frac{dD}{dv_z}$ [mm (m s$^{-1}$)$^{-1}$] is used as a coordinate transformation from diameter to velocity,
where $v_z$ (m s$^{-1}$) is the raindrop terminal velocity of diameter D (mm) at altitude z. $\lambda$ is the
wavelength of radar. $|K_w|^2$ equals 0.93 and it is the dielectric factor.
The $N_{DSD}(D)$ can be expressed as a normalized gamma shape distribution with a three
parameters (Leinonen et al., 2012):
$$N_{DSD}(D; N_w, D_m, \mu) = N_w f(D; D_m, \mu),$$
(A2)

where
$$f(D; D_m, \mu) = \frac{6}{4^4} \frac{(\mu+4)^{(\mu+4)}}{\Gamma(\mu+4)} (\frac{D}{D_m})^\mu \exp\left[-(\mu+4)\frac{D}{D_m}\right].$$
(A3)

$N_w$ is the scaling parameter, $\mu$ is a shape parameter, $\Gamma(x)$ is the Euler gamma function, and $D_m$ is
a mean mass-weighted raindrop diameter estimated from the ratio of the fourth to third DSD
moments:
$$D_m = \frac{M_4}{M_3} = \frac{\int_{D_{min}}^{D_{max}} N_{DSD}(D) D^4 dD}{\int_{D_{min}}^{D_{max}} N_{DSD}(D) D^3 dD} .$$
(A4)

where $D_{min}$ and $D_{max}$ represent the minimum and maximum diameters in the distribution,
respectively.

The intrinsic (non-attenuation) reflectivity factor and the mean velocity and the spectrum

variance are the zeroth, first, and second reflectivity-weighted velocity spectrum moments :
$$Z^\lambda_{DSD} = \sum_{v_{min}}^{v_{max}} S^\lambda_{DSD}(v_i) \Delta v$$
(A5)

$$v^\lambda_{DSD} = \frac{\sum_{v_{min}}^{v_{max}} S^\lambda_{DSD}(v_i) v_i \Delta v}{Z^\lambda_{DSD}}$$
(A6)

$$SV^\lambda_{DSD} = \frac{\sum_{v_{min}}^{v_{max}} (v_i - v^\lambda_{DSD})^2 S^\lambda_{DSD}(v_i) \Delta v}{Z^\lambda_{DSD}}.$$
(A7)



where $v_i$ is the discrete velocities and $\Delta v$ is velocity resolution in the integration.
The Doppler Velocity Difference (DVD) is defined as
$$DVD = v_{DSD}^{3\,GHz} - v_{DSD}^{35\,GHz}. \qquad (A8)$$
Note that both DVD and SV are dependent on DSD parameters ($D_m$ and $\mu$) only.

The RLWC and rain rate (RR) can also be described using the DSD:

$$RLWC(g\,m^{-3}) = \frac{\pi}{6} 10^{-3} \sum_{D_{min}}^{D_{max}} N_{DSD}(D, N_w, D_m, \mu) D_i^3 \Delta D \qquad (A9)$$
$$RR(mm\,hr^{-1}) = \frac{6\pi}{10^4} \sum_{D_{min}}^{D_{max}} N_{DSD}(D, N_w, D_m, \mu) D_i^3 v_z(D_i) \Delta D. \qquad (A10)$$
In addition, there are two newly defined radar-related parameters ($Z_{3GHZ}LWC$ and $Z_{3GHZ}RR$),
which are also dependent on $D_m$ and $\mu$ only:
$$Z_{3GHZ}LWC = 10 \log_{10}(Z_{DSD}^{3GHz}/RLWC) \qquad (A11)$$
$$Z_{3GHZ}RR = 10 \log_{10}(Z_{DSD}^{3GHz}/RR) \qquad (A12)$$

In this study, four variables, DVD, SV at 35 GHz ($SV_{35GHz}$), $Z_{3GHZ}LWC$ and $Z_{3GHZ}RR$, are

pre-calculated using different groups of $D_m$ and $\mu$ values, and then these values are stored in
look-up tables (LUTs). Raindrop backscattering cross sections are calculated using the T-matrix
with different temperatures and oblate raindrop axis ratios (Leinonen, 2014). LUT examples are
illustrated in Fig. A as functions of DVD and $SV_{35GHz}$. If we assume that the observed radar
Doppler velocity difference and spectrum variance from the 35-GHz radar is equal to the DSD
velocity difference and variance (DVD and $SV_{35GHz}$), the measured Doppler velocity difference
and spectrum variance at 35-GHz can determine a solution for $D_m$ from the LUT (Fig. A(a)).
Similarly, a value of $Z_{3GHZ}LWC$ (or $Z_{3GHZ}RR$) can be found with measured DVD and $SV_{35GHz}$
using the LUT in Fig. A(b) (or Fig. A(c)). Then RLWC (or RR) can be estimated using (A11)
(or (A12)) with measured reflectivity at 3-GHz ($Z_{3GHZ}$).





The observed radar Doppler velocity difference can be assumed to be equal to the DSD
velocity difference for two reasons: (1) even though the radar observed Doppler velocity
spectrum can be broaden by the air motion, this spectrum broadening variance is small (within
2%) relative to the DSD velocity spectrum because of the narrow beamwidth (0.2º) of KAZR
and (2) spectrum broadening is symmetric, which does not affect the first spectrum moment and
the DSD mean Doppler velocity only shifts due to the air motion.  Therefore, the measured
differences of Doppler velocity between the 3-GHz and 35- GHz radars vertical pointing
observations are independent of air motion and can be assumed to be the same as DVD from
(A8).  The validity of such an assumption is fully discussed in Williams et al. (2016).
The variabilities of 3-GHz and 35-GHz VPR observations within each 1-minute/60-meter
bin are regarded as the measurement uncertainties and will be propagated through the retrieval to
produce retrieval uncertainties.  The retrieval uncertainties are estimated follow two steps: (1)
construct a distribution of input radar measurements.  For example, the temporal resolution for 3-
GHz VPR is seven seconds, thus there are about nine radar reflectivities observed for one minute.
A normal distribution is generated first using the mean and standard deviations of these nine
observed radar reflectivities for this 1-min/60-m resolution/bin.  (2) repeat the DVD retrievals
using samplings from distributions of all input measurements.  We randomly select 100 groups
of members from those (DVD, $SV_{35GHz}$, $Z_{3GHZ}$) normal distributions to form 100 realizations, and
then produce 100 separate output estimates.  The mean and standard deviation of the 100
solutions are regarded as the final retrieval and the retrieval uncertainty.
It is noted that the uncertainty here only considers estimates of instrument noise, not the
uncertainties associated with assumptions used in the retrieval.  For example, the gamma size
distribution used in (A2) is an approximation which may introduce error into the retrieval.





However, it is very difficult to quantify this type of retrieval uncertainty. In this study, we
further compared our retrievals with independent surface disdrometers measurements to estimate
the uncertainties of retrievals. Also, when both radars are observing at Rayleigh scattering for
small raindrops, the reflectivity-weighted radial velocities for these particles should be the same.
In order to have a difference in radial velocity during the retrieval, large droplets must exist. The
maximum diameters in drop size distribution measured from disdrometer for all the stratiform
cases during MC3E are investigated. It is found that the occurrence of small-droplets-only
(maximum diameter <1.3 mm) is very low (less than 3%). Thus, it will not have a significant
impact on the retrieval results. Notice that this algorithm is not suitable for strong convective
rain due to the wind shear and strong turbulence as well as severe attenuation and extinction of
the Ka-band radar signal.

**Acknowledgments:** J. Tian and X. Dong are supported by DOE CMDV project under grant DE-
SC0017015 at the University of Arizona, and B. Xi is supported by NASA CERES project under
grant NNX17AC52G at the University of Arizona. C. R. Williams is supported by DOE ASR
project under grant DE-SC0014294. Special thanks to Dr. Sergey Matrosov from NOAA Earth
System Research Laboratory (ESRL) for his suggestions. Special thanks to Michael Jensen, PI
of MC3E. Aircraft in situ measurements are processed using data from
https://ghrc.nsstc.nasa.gov/pub/fieldCampaigns/gpmValidation/mc3e/, can also be obtained from
Xiquan Dong (xdong@email.arizona.edu). NOAA vertical profile radar datasets are publically
available in the DOE archives (http://iop.archive.arm.gov/arm-iop/2011/sgp/mc3e/williams-
s_band/).



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



**Table 1.** Acronyms and Abbreviations Used in This Study

| Acronyms and Abbreviations | Full Name |
| --- | --- |
| 2DVD | Two-dimensional video disdrometer |
| A | Total two-way attenuation of 35-GHz VPR signals |
| ARSCL | Active remote sensing of clouds |
| ARM | Atmospheric Radiation Measurement |
| B | coefficients for cloud water attenuation |
| C | coefficients for rainfall attenuation |
| CLWP | Cloud liquid water path |
| D | Raindrop diameter |
| $D_m$ | Mean mass-weighted raindrop diameter |
| $D_{max}$ | Maximum diameters in the size distribution |
| $D_{min}$ | Minimum diameters in the size distribution |
| DOE | Department of Energy |
| DSD | Drop size distribution |
| DVD | Doppler velocity difference |
| G | Two-way gaseous absorption |
| IWC | Ice water content |
| KAZR | Ka-band ARM zenith radar |
| LUT | Looking up table |
| LWP | Liquid water path |
| MB | Base of melting layer |
| MC3E | Mid-latitude continental convective clouds experiment |
| MMCR | Millimeter-wavelength cloud radar |
| MWR | Microwave radiometer |
| $N_{DSD}$ | Number concentration |
| $N_0$ | Intercept of ice particle size distribution |
| NOAA | National Oceanic and Atmospheric Administration |
| $N_w$ | Scaling parameter in the drop size distribution |
| RLWP | Rain liquid water path |
| $R_m$ | Layer-mean rain rate |
| RR | Rain rate |



| Symbol | Description |
|---|---|
| $S_{DSD}^\lambda$ | Radar reflectivity-weighted velocity spectral density |
| $v_{DSD}^\lambda$ | First reflectivity-weighted velocity spectrum moments |
| | represent the mean velocity |
| Vz | Raindrop terminal velocity |
| $Z_{DSD}^\lambda$ | Zeroth reflectivity-weighted velocity spectrum moments |
| | represent the intrinsic (non-attenuation) reflectivity factor |
| $\Gamma(\mathbf{x})$ | Euler gamma function |
| $\lambda$ | Radar wavelength |
| $\sigma_b^\lambda$ | Raindrop backscattering cross section |
| $\mu$ | Shape parameter |





**Table 2.** Statistics (mean differences, 95% confidence interval of mean differences, RMSEs) of $D_m$, RR between this study (RET) and disdrometers (2DVD, RD-80) for Case A and Case B

| | Mean Differences (95% confidence interval) | RMSE |
|---|---|---|
| Case A: $D_m$ (RET, 2DVD) (mm) | -0.02 (-0.05, 0.01) | 0.24 |
| Case A: RR (RET, RD-80) (mm hr⁻¹) | -0.29 (-0.40, -0.19) | 0.98 |
| Case A: RR (RET, 2DVD) (mm hr⁻¹) | -0.31 (-0.48, -0.15) | 1.45 |
| Case B: $D_m$ (RET, 2DVD) (mm) | -0.11 (-0.14, -0.07) | 0.29 |
| Case B: $D_m$ (RET, 2DVD-all) (mm) | -0.09 (-0.13, -0.05) | 0.34 |
| Case B: RR (RET, RD-80) (mm hr⁻¹) | 0.34 (0.16,0.53) | 1.63 |
| Case B: RR (RET, 2DVD) (mm hr⁻¹) | 0.17(-0.01,0.36) | 1.61 |
| Case B: RR (RET, 2DVD-all) (mm hr⁻¹) | 0.14 (-0.08,0.36) | 1.89 |




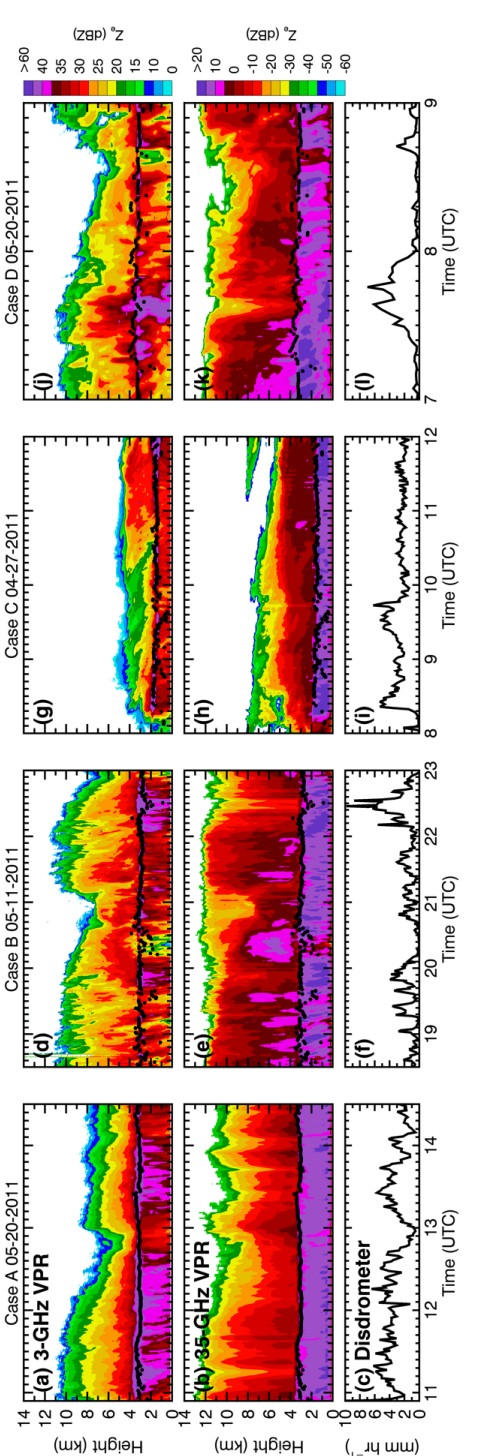

**Figure 1.** Time series of (a) radar reflectivity ($Z_e$) from NOAA 3-GHz vertical pointing radar (VPR), (b) radar reflectivity from ARM 35-GHz VPR, and (c) rain rates from RD-80 surface disdrometer measurement for Case A (20 May 2011, 11:00 – 15 :30 UTC); (d)-(f) for Case B (11 May 201, 18:30 – 23 :00 UTC); (g)-(i) for Case C (27 April 2011, 8:00 – 12 :00 UTC); (j)-(l) for Case D (20 May 2011, 7:00 – 9 :00 UTC). Ceilometer cloud base height estimates are shown with black dots at 1-minute resolution. Note that the ranges of radar dBZ values are different in 3-GHz and 35-GHz radars.







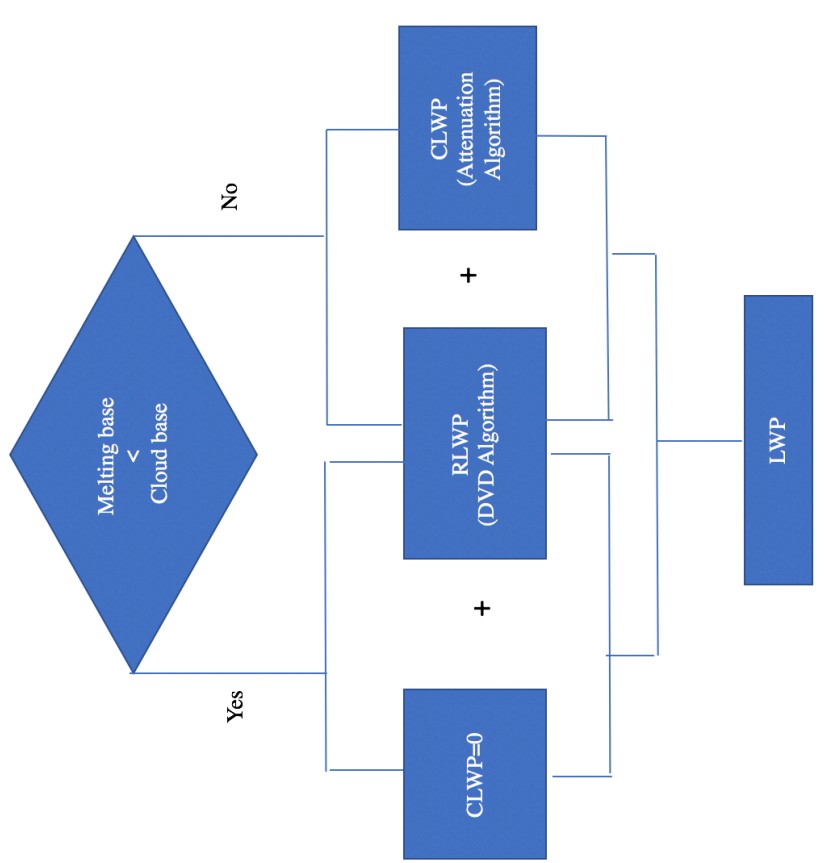

**Figure 2.** Algorithm flowchart to retrieve liquid water path (LWP) below melting base.







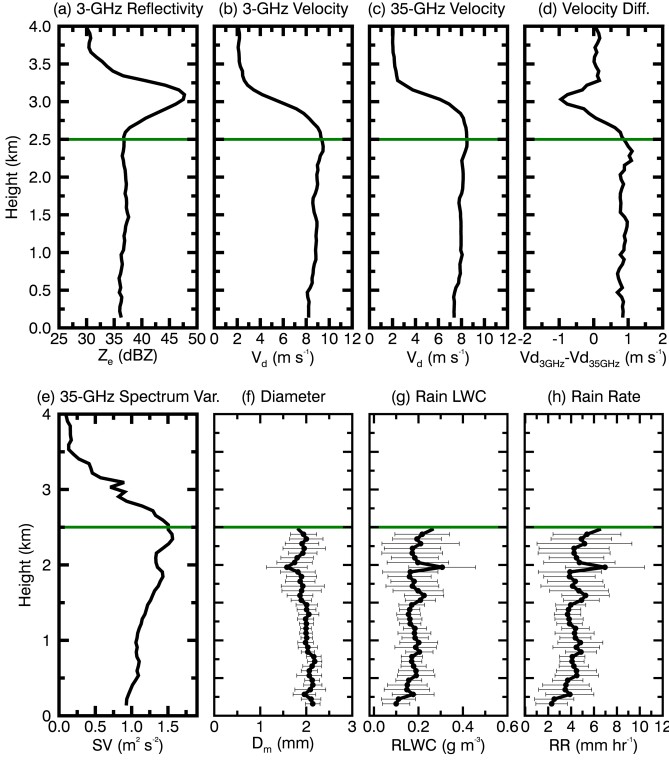


**Figure 3**. An example of illustrating the Doppler Velocity Differences (DVD) retrieval
algorithm at 13:40 UTC on May 20, 2011. The inputs of the DVD retrieval algorithm are: (a) 3-
GHz vertical pointing radar reflectivity factor ($Z_e$), (b) 3-GHz radar Doppler velocities ($V_d$), (c)
35-GHz radar Doppler velocities ($V_d$), and (e) 35-GHz radar spectrum variances (SV). The
Doppler velocity difference between 3-GHz and 35 GHz is shown in (d). The outputs of the
DVD retrieval algorithm are: (f) mass-weighted mean diameter $D_m$, (g) rain liquid water content
(RLWC), and (h) rain rate (RR). Retrieval uncertainties are shown as horizontal thin black lines.

679




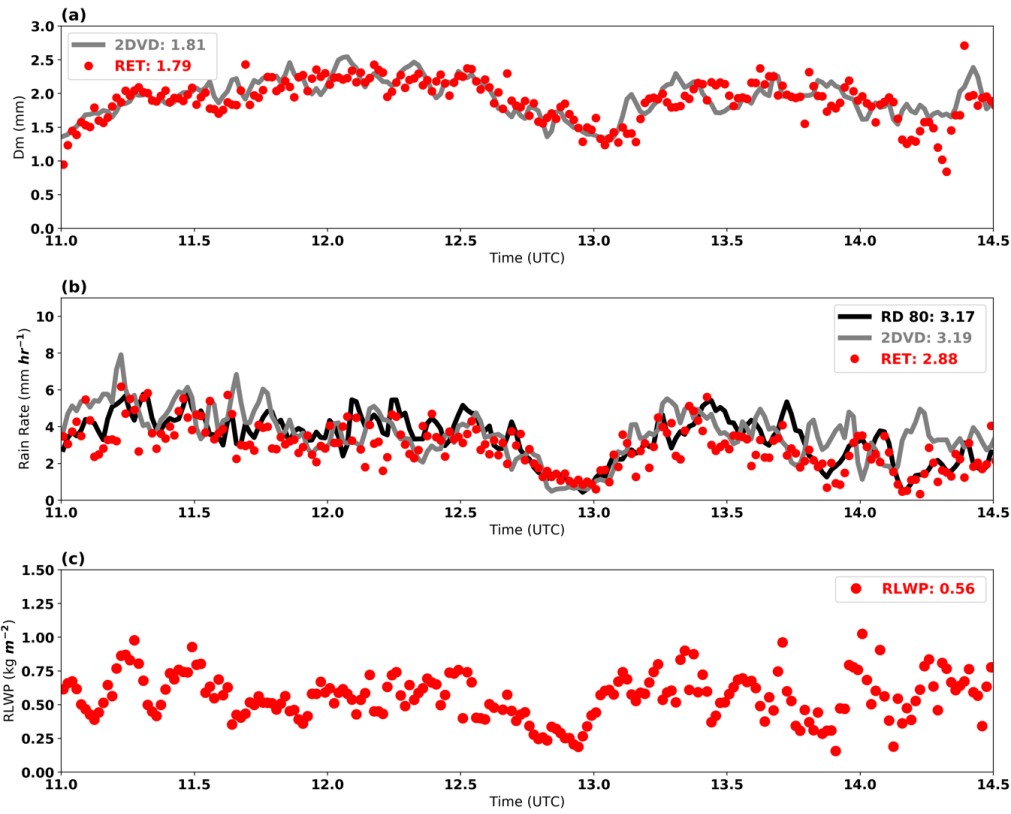

680

**Figure 4.** Time series of (a) retrieved (RET) (red dots) and 2DVD surface disdrometer estimated
(grey line) $D_m$, (b) RET (red dots), 2DVD (grey line) and RD-80 (black line) surface disdrometer
rain rate estimates, and (c) retrieved rain liquid water path (RLWP, red dots) for Case A (May 20,
2011.

685



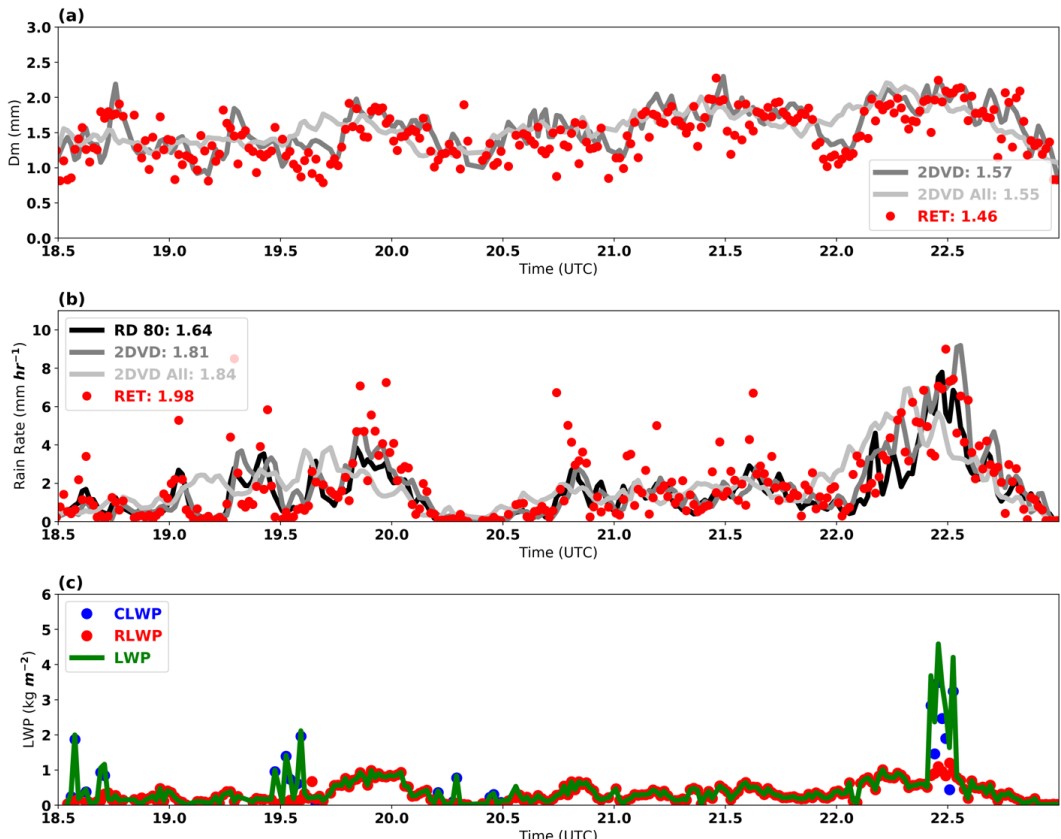

686

**Figure 5.** Time series of (a) retrieved (RET) (red dots) and 2DVD surface disdrometer estimated
(grey lines) $D_m$, (b) RET (red dots), 2DVD (grey line) and RD-80 (black line) surface
disdrometer rain rate estimates, and (c) rain liquid water path (RLWP, red dots), cloud liquid
water path (CLWP, blue dots) and liquid water path (LWP = RLWP+CLWP, green lines) for
Case B (May 11, 2011).




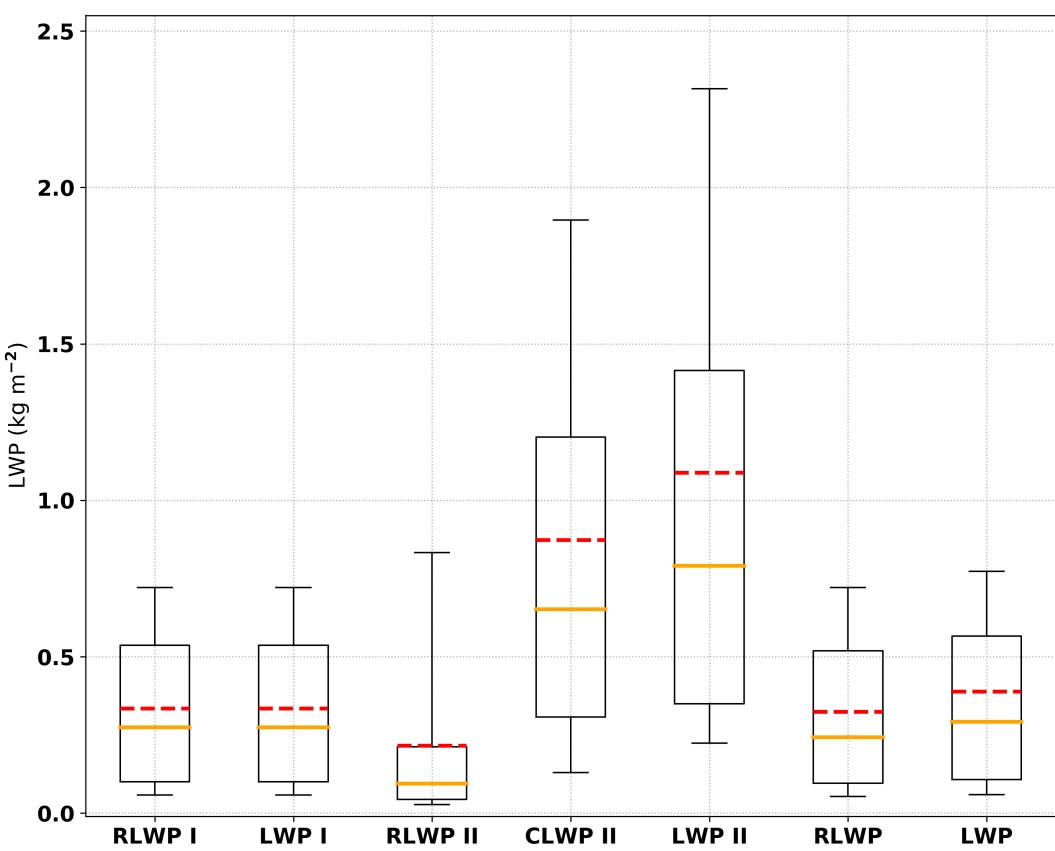

**Figure 6**. Box and whisker plots of retrieved RLWP, CLWP and LWP for situation (I), (II) and
all samples. The horizontal orange line within the box indicates the median, boundaries of the
box indicate the 25th- and 75th -percentile, and the whiskers indicate the 10th- and 90th -percentile
values of the results. The red dash lines indicate the mean values.






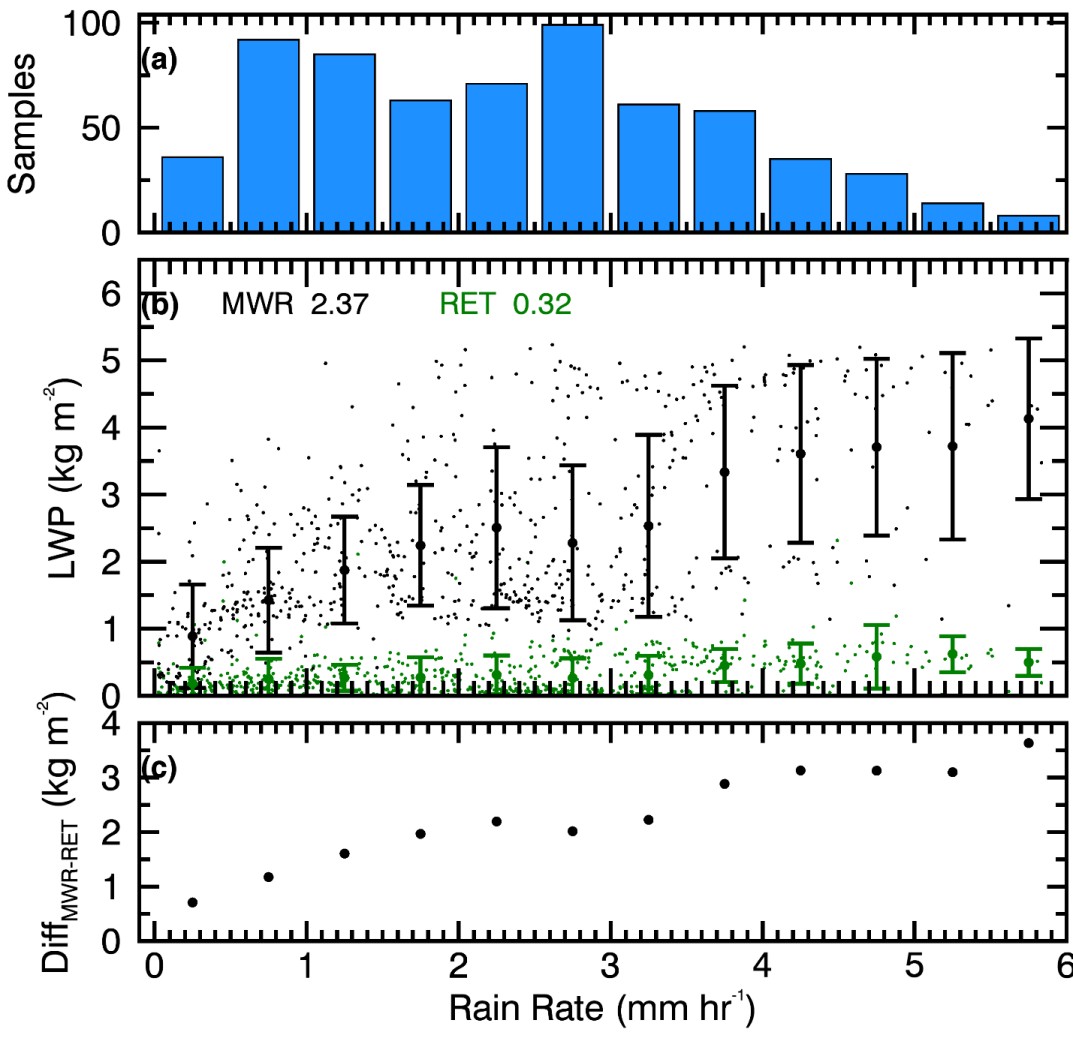


**Figure 7**. (b) Statistic comparisons between LWP retrievals from this study (RET, dots with one
standard deviation bars in green) and microwave radiometer (MWR, black dots with one
standard deviation bars in black), (a) corresponding sample numbers (blue bars) in each rain rate
bin (0.25 mm hr⁻¹), and (c) the LWP differences between two estimations, shown as a function of
rain rate for all cases.

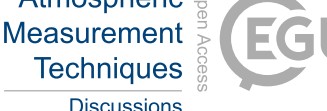

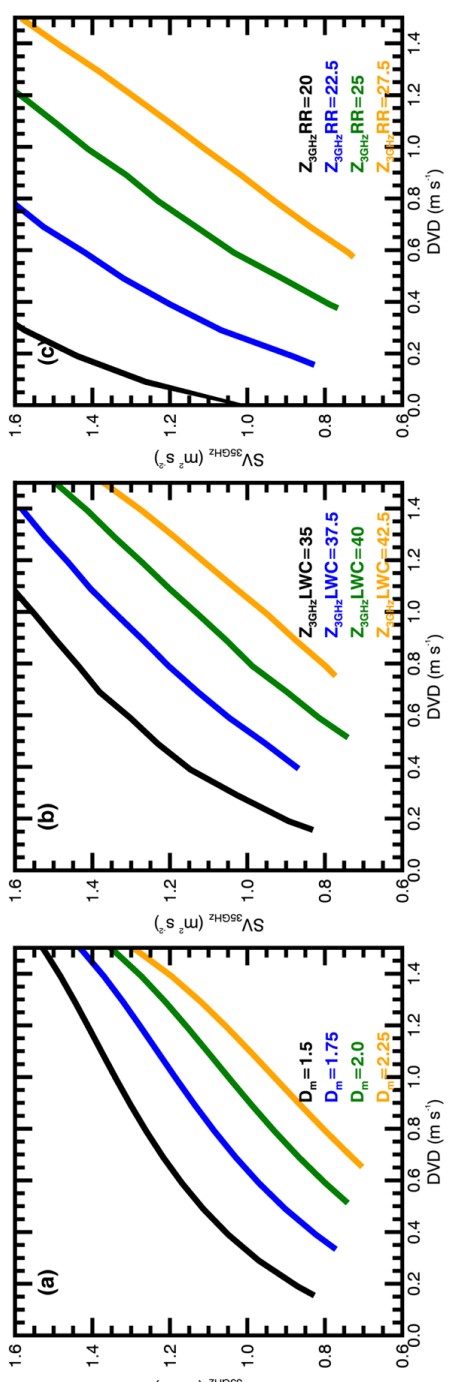

**Figure A.** Comparisons of (a) mass-weighted mean diameter $D_m$ (mm), (b) parameter $Z_{3GHz}LWC = 10 \log(Z_{3GHz}/LWC)$ (dB), and (c) parameter $Z_{3GHz}PR = 10 \log(Z_{3GHz}/RR)$ (dB) calculated as functions of Doppler velocity difference (DVD) and spectrum variance at 35 GHz ($SV_{35GHz}$).
