# Peer review of "Estimation of Liquid Water Path in Stratiform Precipitation Systems using Radar Measurements during MC3E"

_Atmospheric Measurement Techniques, 2018_

## Referee Comment (RC1) · Anonymous Referee #1 · 21 Feb 2019

See the supplement.

Please also note the supplement to this comment:
https://www.atmos-meas-tech-discuss.net/amt-2018-388/amt-2018-388-RC1-supplement.pdf

---

## Author Comment (AC1) · 25 Apr 2019

The comment was uploaded in the form of a supplement:
https://www.atmos-meas-tech-discuss.net/amt-2018-388/amt-2018-388-AC1-supplement.pdf

---

## Author Response (AR1)

*Review: Estimation of Liquid Water Path in Stratiform Precipitation Systems using Radar Measurements during MC3E*

*By Authors: Jingjing Tian, Xiquan Dong, Baike Xi, Christopher R. Williams, and Peng Wu*

*General comments:*

*The authors present a scheme to retrieve rain liquid water path (RLWP) and cloud liquid water path (CLWP) beneath the melting layer in stratiform precipitation. It is known that LWP retrieved from traditional microwave radiometer (MWR) in rainy conditions might be invalid. The authors consider two situations: (1) no cloud detected below the melting layer; (2) cloud is detected below the melting layer. The retrieval of RLWP (namely, situation 1) is based on the method proposed by Williams et al. (2016). In situation (2), to retrieve CLWP they estimate the layer mean rain rate by using the differential velocity technique proposed by Williams et al. (2016), which is expected to be an improvement to the original method (Matrosov. 2009). Using the proposed technique, the authors found that the CLWP is the main contributor to LWP beneath the melting layer when cloud exists.*

*This is an interesting study that is relevant to the scientific community and is within the scope of AMT. Overall, this manuscript is well written, cites relevant literature and the proposed technique is novel. However, the authors should include a more detailed discussion about the retrieval uncertainty. I recommend it for publication in AMT if the authors take into account the following comments.*

**We would like to thank this reviewer for his/her time in reviewing this manuscript and providing very clear and insightful suggestions.**

**Based on this reviewer's suggestion, we made some efforts in the following aspects:**
**(1) Carefully checked datasets from different observations, which include the ceilometer detected cloud base, the melting base identified from radar observations, for every data point one-by-one.**
**(2) Added more uncertainty analyses for retrievals. More details please see the response for the second major comment.**
**(3) Checked if there are any relationships between retrievals and the melting base height.**

**The reviewer's comments really led us to think deeper of our own study and help us a lot to improve the manuscript through answering the reviewer's questions. This is greatly appreciated. The responses are in bold and black.**

**We also attached the revised manuscript with track changes at the end of the response to review.**

*Major comments:*

*1) Given the generally recognized definition of liquid water path (LWP) is the integral of the liquid water in the whole atmospheric column, I suggest the authors highlight that the LWP in this*

*manuscript is beneath the melting base in title, abstract and other places where misunderstanding might be induced.*

**Agree.**
**The title has been changed to "Estimation of Liquid Water Path *below the Melting Layer* in Stratiform Precipitation Systems using Radar Measurements during MC3E"**

**Also, clarifications are added in the abstract and other places.**
**For example, the first sentence of in the abstract changed as "In this study, the liquid water path (LWP) below the melting layer in stratiform precipitation systems is retrieved, which is a combination of rain liquid water path (RLWP) and cloud liquid water path (CLWP). "**

*2) I believe a more detailed retrieval uncertainty analysis for situation (2) is needed. I would love to see the retrieval uncertainty in Figure 4 and 5 (see examples in Figure 3 f, g, h). In addition, a more thorough comparison with (Matrosov, 2009) regarding the retrieved results as well as the uncertainty would be interesting.*

**As suggested, we added the retrieval uncertainties in Figure 4 and Figure 5.**
**Figure 4 shows the retrievals for rain microphysics only. The uncertainties for $D_m$ and rain rate are discussed in our original manuscript (Appendix A). "The uncertainties are estimated using the Monte Carlo method, by first estimating radar measurement uncertainties, constructing a distribution of input radar measurements and then repeating the DVD retrievals for this distribution of input measurements. The details of this "DVD Algorithm" and uncertainty estimation are introduced in Appendix A."**

**The uncertainties of RLWP (Figure 4c) is estimated based on the uncertainties of RLWC. More specifically, step (1) we estimated the RLWC uncertainties at each height level, which is similar to the uncertainty estimations of $D_m$ and rain rate in Appendix A. Step (2), we calculated the ratios of RLWC uncertainties to mean retrieved RLWC at each height level, which represent percentage values of retrieval uncertainties. Step (3), we calculated the mean ratio of the uncertainties in the whole liquid layer below melting base and regarded this mean ratio as the uncertainties of RLWP.**

**In the Appendix A we added "The uncertainties of RLWP are estimated based on the uncertainties of RLWC. More specifically, we first estimated the RLWC uncertainties at each height level, and then we calculated the ratios of RLWC uncertainties to mean retrieved RLWC at each height level, which represent percentage values of retrieval uncertainties. Finally, we calculated the mean ratio of the uncertainties in the whole liquid layer below melting base and regarded this mean ratio as the uncertainty of RLWP."**

**The updated Figure 4 is shown as below:**

[Figure]

**Figure 4.** Time series of (a) retrieved (RET) (red dots) and 2DVD surface disdrometer estimated (grey line) $D_m$, (b) RET (red dots), 2DVD (grey line) and RD-80 (black line) surface disdrometer rain rate estimates, and (c) retrieved rain liquid water path (RLWP, red line) for Case A (May 20, 2011). The red shading areas are the estimated retrieval uncertainties.

**In Figure 5, the estimation methods for rain microphysics retrieval uncertainties are the same as those we discussed in Figure 4.**

**The uncertainty of CLWP is discussed as follows:**

**CLWP is estimated by, firstly, subtracting the total attenuation (A) from rain attenuation to get the cloud attenuation, and secondly, dividing the cloud attenuation (B) by the cloud water attenuation coefficient.**

$$CLWP = \frac{A - 2\,C\,R_{total}}{2\,B} \tag{1}$$

**The rain attenuation is estimated by the rain attenuation coefficient (C) multiplied by the total rain rate ($R_{total}$). The attenuation (A) is estimated by comparing the drop in Ka-band reflectivity with the un-attenuated S-band reflectivity. C and B are the coefficients for rain and cloud water attenuation, where C equals ~ 0.26 dB /km /mm hr[-1], and B equals ~ 0.87 dB / kg m[-2]. The influence of temperature uncertainties in B on the retrieval errors is minor**

compared to the uncertainty of the total attenuation (A) and total rain rate ($R_{total}$) (Matrosov 2010).

The uncertainty of CLWP is calculated as

$$\Delta CLWP = \sqrt{(\frac{\partial CLWP}{\partial A} \times \Delta A)^2 + (\frac{\partial CLWP}{\partial R_{total}} \times \Delta R_{total})^2} \qquad (2)$$

$$\Delta CLWP = \sqrt{(\frac{1}{2B} \times A \times Ua)^2 + (-\frac{C}{B} \times R_{total} \times Ur)^2} \qquad (3)$$

For given uncertainties of attenuation (Ua) and total rain rate (Ur), the uncertainty of CLWP can be calculated using equation (3). The blue uncertainty bars in Figure 5c show the retrieved CLWP uncertainty with assuming both of the uncertainties of attenuation and total rain rate are 30% (Ua=Ur=30%). It is noted that, due to the variations of the attenuation and total rain rate with time, the estimated uncertainty of CLWP varies point to point.

[Figure]

**Figure 5.** Time series of (a) retrieved (RET) (red dots) and 2DVD surface disdrometer estimated (grey lines) $D_m$, (b) RET (red dots), 2DVD (grey line) and RD-80 (black line) surface disdrometer rain rate estimates, and (c) rain liquid water path (RLWP, red line), cloud liquid water path (CLWP, blue dots) and liquid water path (LWP = RLWP+CLWP, green lines) for Case B (May 11, 2011). The red shading area and blue bars are the estimated retrieval uncertainties for rain microphysical properties ($D_m$, rain rate and RLWP) and CLWP.

In the revision, we added an Appendix B:
"Appendix B: CLWP Uncertainty
    CLWP can be estimated as following equation:

$$\text{CLWP} = \frac{A - 2\,C\,R_{total}}{2\,B} \; . \tag{B1}$$

The rain attenuation is estimated by the rain attenuation coefficient (C) multiplied by the total rain rate ($R_{total}$). The attenuation (A) is estimated by comparing the drop in Ka-band reflectivity with the un-attenuated S-band reflectivity. C and B are the coefficients of rain and cloud water attenuation with values of ~ 0.26 dB /km /mm hr$^{-1}$ and ~ 0.87 dB / kg m$^{-2}$, respectively. The influence of temperature uncertainty in B on the retrieval error is minor compared to the uncertainties of the total attenuation (A) and total rain rate ($R_{total}$) (Matrosov 2010). The uncertainty of CLWP is calculated as

$$\Delta CLWP = \sqrt{\left(\frac{\partial CLWP}{\partial A} \times \Delta A\right)^2 + \left(\frac{\partial CLWP}{\partial R_{total}} \times \Delta R_{total}\right)^2} \tag{B2}$$

$$\Delta CLWP = \sqrt{\left(\frac{1}{2B} \times A \times \text{Ua}\right)^2 + \left(-\frac{C}{B} \times R_{total} \times \text{Ur}\right)^2} \tag{B3}$$

For given uncertainties of attenuation (Ua) and total rain rate (Ur), the uncertainty of CLWP can be calculated based on equation (B3). "

*In Figure 5c, the retrieved CLWP values can be as large as 2~3 kg/m2. It sounds like very large values for me. Since the comparison to MWR LWP is a suspect due to radome wetting, could you compare your retrieved values to observations in non-precipitating warm clouds? It would give an idea of how realistic those values are.*

To figure out why we have large CLWP values (e.g., ~ 2- 3 kg m$^{-2}$) in the original manuscript, at first, we checked the melting base and cloud base manually and carefully to make sure we use the "best-estimations" of melting base and cloud base in the retrieval.
The new CLWP retrievals are shown in Figure 5c where all CLWP values are smaller than 1.5 kg m$^{-2}$. Also, as suggested, the retrieval uncertainties are added in the Figure 5. The retrieved CLWP values, especially when taken uncertainties into consideration, are comparable with the estimations of CLWP (Figure 13) in Matrosov (2009). It is clearly seen that the retrieved CLWP values could be larger than 1 kg m$^{-2}$. After the revision, the mean and median values of the retrieved CLWP are 0.56 and 0.45 kg m$^{-2}$ for all cases (Figure 6).

[Figure]

FIG. 13. Times series of LWP retrievals and ceilometer estimates of the cloud-base heights.

[Figure]

**Figure 6. Box and whisker plots of retrieved RLWP, CLWP and LWP for situation (I), (II) and all samples. The horizontal orange line within the box indicates the median, boundaries of the box represent the 25th- and 75th -percentile, and the whiskers indicate the 10th- and 90th -percentile values of the results. The red dash lines represent the mean values.**

*Since the comparison to MWR LWP is a suspect due to radome wetting, could you compare your retrieved values to observations in non-precipitating warm clouds? It would give an idea of how realistic those values are.*

**The algorithms in this study do not work for the non-precipitation warm clouds, as there is almost no observations/reflectivities at 3 GHz radar for low-level warm cloud. Thus, we did not compare our retrieved CLWP with MWR-retrieved CLWP directly.**

**But, to get a better idea of how realistic of retrieved CLWP values, as suggested, we checked the CLWP from MWR for non-precipitating warm clouds. An example of CLWP from MWR on May 13, 2011 is shown as below. Observations (Ka-band cloud radar and MWR-LWP) are from ARM SGP site. We found the CLWPs are less than 0.4 kg m$^{-2}$ in general.**

[Figure]

*3) Figure 1. Have you checked that how often the ceilometer signal is totally attenuated?*

**Yes, we checked the ceilometer signal attenuation rate, which is 11.7 %.**

*The black dots in Figure 1 seem indistinguishable from the melting layer for me. I suggest the authors to use a separate time series plot to show the heights of melting top/base and cloud base, which may help readers to understand the difference between situation 1 and 2 better.*

**Thanks for the suggestion of adding a subplot to show the melting and cloud base to make it clearer to readers. Updated Figure 1 is shown as below.**

[Figure]

**Figure 1.** Time series of (a1) radar reflectivity ($Z_e$) from NOAA 3-GHz vertical pointing radar (VPR), (b1) radar reflectivity from ARM 35-GHz VPR, (c1) melting base (blue lines) and cloud base (black dots), and (d1) rain rates from RD-80 surface disdrometer measurement for Case A (20 May 2011, 11:20 – 14 :30 UTC); (b1)-(b4) for Case B (11 May 201, 18:30 – 22 :00 UTC); (c1)-(c4) for Case C (27 April 2011, 8:30 – 13 :00 UTC); (d1)-(d4) for Case D (20 May 2011, 7:00 – 9 :00 UTC). Note that the ranges of radar dBZ values are different in 3-GHz and 35-GHz radars.

*4) Both RLWP and CLWP are dependent on the geometrical thickness of rain and cloud layers, have you considered the changes of the melting layer height? If there are significant changes of melting layer height, RLWP will be affected.*

**To better answer this question, we generate the following figure.**

**In the upper panel, the x-axis is the melting base height (the height we integrate RLWC), and the y-axis is the RLWP. The data are separated for each case using different colors. Case C (in cyan color) has lower melting base heights and smaller RLWP compared to other cases. We agree that, in general, the retrieved RLWP increases with the increased melting base. However, there are some exceptions. For example, some RLWP values for Case D are much lower (< 0.25 kg m-2) at higher melting bases (2.5-3 km), which warrants a further study.**

**In addition to investigate the RLWP, we also checked the mean RLWC for each case (lower panel). Even though the RLWP differences are large between case C and other cases, the RLWC differences are not that significant compared to the RLWP differences.**

[Figure]

*5) Figure 7. The retrieved LWP in Figure 7b seems to be weakly linked to rain rate, though high correlation between the rain rate and RLWP is shown in Figure 4 and 5. Can you explain this?*

**Based on another comment/suggestion given by this reviewer, "Have you considered using a scatter plot to show the relations between RET and MWR with rain rate indicated by color?", we generated a figure as below.**

[Figure]

**This figure shows the comparisons between LWPs retrieved from microwave radiometer measured brightness temperatures (MWR, in x-axis) and LWPs retrieved from this study (RET, in y-axis, with estimated uncertainty in gray lines). The rain rates are indicated by colors. The black line is 1:1 line.**

**In the revised plot, we do find that the retrieved LWPs are slightly correlating rate rates, but not as strong as the MWR-retrieved LWPs.**

In the revision: "The corresponding LWP uncertainties are also provided as the grey error bar for each retrieval with rain rate indicated by colors. The MWR-retrieved LWPs increase with increased rain rate, and much larger than the new LWP retrievals at high rate rates. The newly retrieved LWPs weakly correlate with rain rates, and most values are less than 1.0 kg m⁻², especially at high rain rates."

*Have you tried to quantify by how much the MWR overestimates the LWP for cases with the similar melting base height?*

**Yes, we were trying to find if the "overestimations" from MWR is highly related to the melting base height, but unfortunately, we think it is hard to quantify or give a general relationship, based on these scattered points as shown below.**

[Figure]

*6) The authors compare their finding to the study of Lebsock et al. (2011). However, they study warm clouds while the presented research focuses on cold cloud precipitation. To what extent this comparison is meaningful?*

**The ratio of CLWP/RLWP comparison between cold and warm cloud may be confusing. They are different type of clouds. We deleted the comparisons in the revision.**

*Minor issues:*

*- Please check the references (e.g., line 582, should be 'J. Appl. Meteor. Climatol.'), and follow AMT's requirement on literature format.*

**Yes, references are written in AMT format in the revision.**

**"Matrosov, S. Y.: A method to estimate vertically integrated amounts of cloud ice and liquid and mean rain rate in stratiform precipitation from radar and auxiliary data, J. Appl. Meteor. Climatol., 48, 1398–1410, doi:10.1175/2009JAMC2196.1, 2009"**

*- Figure 3 d. It seems to me that the negative differential velocity in the melting layer is a bit large. Have you checked the matching of the ranges between those two radars?*

**Yes, we checked the radar data matching both temporally and vertically. The negative differential velocity is not as large as -0.8 m/s always. Several examples are given in the figure below.**

[Figure]

*- Figure 4 and 5. Use standard format for time, such as 12:30 instead of 12.5.*

**Yes, modified.**

*- Figure 4b and 5b. It seems that the retrieved rain rate agrees better with more stable precipitation (e.g., 12.7 - 13.1 in Figure 4b). But what are the reasons for those obviously overestimated retrievals in Figure 5b?*

**As suggested by the reviewer, we added the retrieval uncertainties in Figure 4 and Figure 5. Even though the retrievals (red points) are sometimes much larger than the surface observations (grey or black lines) in Figure 5b, we found the observations are within the retrieval uncertainties (shaded area) generally.**

[Figure]

We would also like to address this question on the basis of retrieval mathematically.

In the retrieval, a normal distribution is generated first for each 1-min/60-m resolution radar measurements using their corresponding mean and standard deviations. For example, the temporal resolution for 3-GHz VPR is seven seconds, thus there are about nine radar reflectivities observed for one minute. A normal distribution is generated using the mean and standard deviations of these nine observed radar reflectivities.

Similarly, we generated the normal distributions for Doppler velocity difference and Ka-band spectrum width.

After generating the normal distributions, we randomly select 100 groups of members from those (DVD, $SV_{35GHz}$, $Z_{3GHz}$) normal distributions to form 100 realizations, and then produce 100 separate output estimates. *The final retrieval is the mean of the 100 solutions*, and the retrieval uncertainty is the standard deviation of the 100 solutions.

For the stable precipitation, the standard deviations of inputs are much smaller than those in the unstable precipitation. Thus, the stand deviations of retrievals would be larger in the unstable precipitation, and the differences between observations and mean values of retrievals could be large.

*- Figure 7. It seems to me that the use of error bar is a bit puzzling, since for a rain rate bin the retrieved/measured LWP are under different conditions (e.g., different melting base). Have you considered using a scatter plot to show the relations between RET and MWR with rain rate indicated by color?*

**Figure 7 is changed following the suggestion. This plot is also discussed in the response to reviewer for the former question.**

[Figure]

**Figure 7. (a) Comparisons between LWP from microwave radiometer (MWR, in x-axis) and LWP retrievals from this study (RET, in y-axis, with estimated uncertainty in black bars) with color-coated rain rate values. (b) the LWP differences between two estimations (MWR-RET), shown as a function of rain rate.**

*- Line 39, 'with-cloud'*

**Yes, modified.**

*- Line 49, 'is still'*

**Yes, modified.**

*- Line 62, 'they are known'*

**Yes, modified.**

*- Line 109, '. However,'*

**Yes, modified.**

*- Line 136, 'bright band'*

**Yes, modified.**

*- Line 179, 'with the aid of'*

**Yes, modified.**

*- Line 229, 'limiting'*

**Yes, modified.**

*- Line 235, 'B'*

**Yes, modified.**

*- Line 292, 'relatively'*

**Yes, modified.**

*- Line 339, 'samples'*

**Yes, modified.**

*- Line 347-358, RLWP/CLWP > 2 in Lebsock et al. (2011) while in this study RLWP/CLWP is on average much smaller, how to explain such difference?*

**The ratio of CLWP/RLWP comparison between cold and warm cloud may be confusing. They are different type of clouds. We deleted the comparisons in the revision.**

*- Line 348, 'having not'*

**Yes, modified.**

*- Line 359, 'compared them'*

**Yes, modified.**

*- Line 363, 'possibly due to'*

**Yes, modified.**

*- Line 424, delete either of the 'separately'*

**Yes, modified.**

*The paper describes a technique for estimating the LWP in stratiform precipitation. The methodology is applied to 20 case studies collected during MC3E. The paper is generally clear and well written and targets a very important issue, the partitioning between cloud and rain liquid contents in precipitating clouds (though it finds only a partial solution to it).*

**We appreciate the reviewer for his/her time and effort for reviewing this paper. Especially, the second comment really helps the first author to gain some experiences in radiative transfer calculation and get a deeper understanding on the cloud water path retrievals at (unpolarized/polarized) microwave radiometer channels. We thank him/her for the constructive comments and suggestions. The responses are in bold and black.**
**We also attached the revised manuscript with track changes at the end of the response to review.**

*Some comments 1) It must be clear from the beginning that the methodology is only capable of computing the LWP below the bright band and that the methodology is not applicable in presence of liquid above the melting layer.*

**Agree. The title has been changed to "Estimation of Liquid Water Path *below the Melting Layer* in Stratiform Precipitation Systems using Radar Measurements during MC3E"**
**Also, clarifications are added in abstract and other places.**
**For example, the first sentence of in the abstract changed as "In this study, the liquid water path (LWP) below the melting layer in stratiform precipitation systems is retrieved, which is a combination of rain liquid water path (RLWP) and cloud liquid water path (CLWP). "**

*2) The explanation on the overestimates of LWPs by radiometers is pretty convoluted (at the moment it is a full page, page 17) and needs to be simplified.*

*To better understand this bit in first place you could compute the extinction (and scattering) coefficients like in Fig.10 of your second reference. Clearly raindrops are much more efficient in extinguishing radiation than cloud droplets (but this depends on the size of the raindrops! so I do not agree with the 2/3 statement at line 382). Yes I agree the single scattering albedo also is much larger.*

*Second you could use RT computations (e.g. Eddington or a successive order approximations where you can simplify all equations because for your purpose you can neglect polarization effects and you can assume spherical particles only) to compute the TBs for the two channels used by the radiometers to show the enhancement when r-LWP instead of c-LWP is present. Fig4 of your second reference shows an example of that for 30 degrees elevation angle (here you need to repeat the computation at nadir and for the frequencies of the radiometer). But from that figure it is clear the enhancement in case of rain: compare the TBs e.g. for c-LWP=1 kg/m^2 vs r-LWP=1 kg/m^2.*

**Following the suggestions, we first generated Figure B as the Figure 10a in Battaglia et al. (2009). We calculated the extinction cross section per volume as a function of the drop equivolume diameter for the two frequencies/channels in MWR (23.8 GHz and 31.4 GHZ) with a T-matrix method. It is notice that the extinction cross section increases with increased diameter when the diameters are smaller than 3 mm. This indicates the extinction (cross**

section) for rain drops (diameter > ~ 50 um) is much larger than that for cloud droplets (diameter < ~50 um).

[Figure]

**Figure B. The extinction cross section per volume as a function of the drop equivolume diameter for the two frequencies in MWR (23.8 GHz and 31.4 GHZ).**

Secondly, we also generated Figure C as the Figure 10 b in Battaglia et al. (2009). We calculated the extinction coefficient as a function of RLWC for populations with three different drop size distributions (DSDs). The DSDs are modeled according to the exponential Marshall and Palmer (MP) distribution $N(D) = N_0 e^{-\Lambda D}$. $N_0=8000 \text{ m}^{-3} \text{ mm}^{-1}$. $N_0$ is changed to 4000 and 32000 $\text{m}^{-3} \text{ mm}^{-1}$ to represent thunderstorm and drizzle DSDs. More details of the DSDs please see Battaglia et al. (2009).

Figure C clearly shows the extinctions of cloud and rain also are DSD-dependent. For example, at 31.4 GHz, even though the RLWC is the same, the extinctions are much larger from the precipitation with the thunderstorms and MP DSDs than the extinctions from light precipitation with the drizzle DSD.

[Figure]

**Figure C. The extinction coefficient as a function of RLWC for precipitations with three different drop size distributions (DSDs) in which they represent heavy precipitation (thunderstorm), moderate precipitation (M&P) and drizzle precipitation (drizzle).**

[revised manuscript text omitted]

*3)Also a key effect in enhancing brightness temperature is the presence of the melting layer (relevant literature must be cited).*

Yes, Battaglia et al (2003) found the brightness temperature generally increases if mixed-phase precipitation is included.

In the revision, "The LWP differences between MWR retrieval and this study could be caused by the following reasons. 1) MWR-retrieved LWP represents the entire vertical column (RWLP and CLWP below melting layer, large water coated ice particles in the melting layer and supercooled LWCs above the melting layer), while our retrieval only represents the LWP below the melting base.  As Battaglia et al (2003) pointed out the brightness temperature generally increases if mixed-phase precipitation is included. 2) The MWR radome was wet during the raining periods and the deposition of raindrops on the radome can cause a large increase in the measured brightness temperatures (Cadeddu et al., 2017). 3) Large extinctions due to rain drops would affect MWR retrievals.  4) Uncertainties exist in the retrieved LWP from this study."

*4)I found also the discussion at line 315-324 a bit confused: I am not sure why you want to include other disdrometers located within 5 km. I would suggest to delete it.*

**As suggested, we only include the comparison from the closed 2DVD and RD-80 with retrievals. The comparisons between measurements from other 2DVDs and the corresponding discussions are deleted.**

*5) Figure A: it would be good to see also contour lines with the values of mu.*

**As suggested, the contour lines with the miu values are added in the Figure A (b).**

[Figure]

**Figure A. Comparisons of (a) mass-weighted mean diameter $D_m$ (mm), (b) shape parameter $\mu$, (c) parameter $\alpha = 10 \log(Z_{3GHz}/RLWC)$, and (d) parameter $\beta = 10 \log(Z_{3GHz}/RR)$ calculated as functions of Doppler velocity difference (DVD) and spectrum variance at 35 GHz ($SV_{35GHz}$). Note that the units of RLWC and RR are g m$^{-3}$ and mm hr$^{-1}$.**

*6) Several typos (e.g. line 196, 292)*

**Thanks for the carefully check. The typos are corrected.**

*7) The names of the parameters in the Appendix are not optimal, e.g. Z_3GHz RR does not suggest a ratio. Also their units is not dB as stated in the caption of Fig.1. (dB corresponds to 10 log10 of a UNITLESS quantity!!!); here you are defining a very specific unit (like the dBZ, you need to specify the units used for z and LWC).*

**Variable names are changed.**

In the revision, the variables are defined as $\alpha$ and $\beta$

$$\alpha=10 \log_{10}(Z_{DSD}^{3GHz}/RLWC) \tag{A11}$$

$$\beta=10 \log_{10}(Z_{DSD}^{3GHz}/RR) \tag{A12}$$

The review's comments are right, the units of $\alpha$ and $\beta$ should not be dB.
Instead of defining the units of $\alpha$ and $\beta$, as suggested, the units of RLWC and rain rate are specified in the captions of Figure A.

[revised manuscript text omitted]

EXT COEFF($km^{-1}$)

RLWC (g $m^{-3}$)

Thurderstorm 31.4 GHz
MP 31.4 GHz
Drizzle 31.4 GHz
Thurderstorm 23.8 GHz
MP 23.8 GHz
Drizzle 23.8 GHz

[Figure]

**Figure C.** The extinction coefficient as a function of RLWC for precipitations with three different drop size distributions (DSDs), which are for heavy precipitation (thunderstorm), moderate precipitation (MP) and drizzle precipitation (drizzle).

| Page 34: [1] Deleted | Tian, Jingjing - (jingjingtian) | 4/19/19 12:47:00 AM |

| Page 34: [2] Deleted | Tian, Jingjing - (jingjingtian) | 4/19/19 12:47:00 AM |